# Individual Determinants as the Causes of Failure in Learning to Swim with the Example of 10-Year-Old Children

**DOI:** 10.3390/ijerph19095663

**Published:** 2022-05-06

**Authors:** Andrzej Ostrowski, Arkadiusz Stanula, Andrzej Swinarew, Alexander Skaliy, Dariusz Skalski, Wojciech Wiesner, Dorota Ambroży, Krzysztof Kaganek, Łukasz Rydzik, Tadeusz Ambroży

**Affiliations:** 1Department of Water Sports, Academy of Physical Education, 31-571 Krakow, Poland; andrzej.ostrowski@awf.krakow.pl; 2Institute of Sport Sciences, Jerzy Kukuczka Academy of Physical Education, Mikołowska 72a, 40-065 Katowice, Poland; andrzej.swinarew@gmail.com; 3Institute of Materials Science, University of Silesia in Katowice, 40-007 Katowice, Poland; 4Institute of Sport and Physical Culture, University of Economy, 03057 Kyiv, Ukraine; skaliy@wp.pl; 5Department of Swimming and Water Rescue, Lviv State University of Physical Culture, 79000 Lviv, Ukraine; dariusz.skalski@awf.gda.pl; 6Faculty of Recreation, Academy of Physical Education in Wrocław, 51-612 Wroclaw, Poland; wojciech.wiesner@awf.wroc.pl; 7Institute of Sport Sciences, University of Physical Education in Krakow, 31-571 Krakow, Poland; dorota.ambrozy@awf.krakow.pl (D.A.); tadek@ambrozy.pl (T.A.); 8Department of Coaching and Innovation, Faculty of Tourism and Recreation, Institute of Entrepreneurship and Management, University of Physical Education in Cracow, 31-571 Krakow, Poland; krzysztof.kaganek@awf.krakow.pl

**Keywords:** swimming, teaching, fear of water, morphological characteristics, functional characteristics, coordination motor abilities

## Abstract

Background: The purpose of the present study was to identify which, and to what extent, selected individual determinants of 10-year-old children may limit the final achievement in learning to swim. In view of the above, the research hypothesis was formulated that some children, despite regular attendance at swimming classes, do not achieve the learning outcomes set in the curriculum. The reason for this may be unfavorable (compared to their peers) morphological and functional characteristics, coordination motor abilities, and problems with fear of water. Methods: The study was conducted on a group of 271 students from the third grade of elementary schools who could not swim when they entered the physical education classes at the swimming pool and then participated in at least 25 swimming lessons during the school year. After these classes, the students performed swimming tests, and their somatic and functional characteristics and coordination motor abilities were measured. Results: In 46.1% of the participants, the final achievement level was lower than assumed in the school curriculum. The biggest problem for teachers and students in the initial teaching and learning to swim was the high fear of water, especially among girls. Furthermore, children characterized by lower body height and body weight, a lower sum of three skinfolds, and lower BMI had problems with progress in swimming. Despite the differences, these values did not correlate significantly with the final achievement level in swimming, except for body height in boys. Slower progress in swimming was also associated with lower vital capacity, whereas no relationship was found between final achievement level in swimming and trunk flexibility or foot mobility. However, significant correlations occurred for coordination motor abilities, as in almost all tests the participants characterized by the achievement level below the objectives set out in the curriculum performed significantly worse than children in the group with the achievement level meeting the objectives. Conclusions: In many cases, children who begin learning to swim from scratch make significant progress, but for many of them, the achievement levels are lower than the requirements set out in the school curriculum. The biggest problem for teachers and students in the initial teaching and learning to swim was the high fear of water, especially among girls.

## 1. Introduction

Performing certain movements in the water improves the ability of a swimmer to float on the surface of the water. Learning to swim is a special case in which changes in motor behavior in water, resulting in floating and locomotion, occur as a result of a variety of exercises that familiarize individuals with water resistance and buoyant force [1,2,3,4]. The ability to swim provides a foundation for safety in the aquatic environment and is integral to psychological well-being, development, and maintenance of overall fitness [5,6,7,8]. This ability is also a cultural achievement, which is why in many European countries swimming has been included in physical education programs, especially in elementary schools [9,10,11].

The assessment of the effectiveness of learning to swim and improvements in swimming skills is a difficult issue because there is a paucity of unambiguous criteria, whereas the determinants of learning, although complementary, may be different and variable in the course of learning. Therefore, specific individual determinants may be an advantage for some students, while they may be a hindrance for others. For this reason, some students are able to meet the curriculum objectives and achieve the assumed levels of skills and competencies in swimming, but there are others who fail to do so [12]. Resistance to infections, current health status, and previous negative experiences in water are also important [13,14,15]. Significant facilitators or impediments to learning to swim have also been attributed to body buoyancy, swimming skills, water feeling, and motivation [16,17,18].

It is considered that the factors with the greatest effect on the final achievement level in swimming, especially in the early stages, include age, gender and related physical, motor, mental and social development, and environmental conditions such as the teacher and the methods, forms, and means of teaching they use [2,19,20,21,22,23]. Participation in an adequate number of swimming lessons, regardless of age and gender, is fundamental to acquiring swimming skills [24,25,26].

Learning to swim is best started between 4 and 6 years of age, while the ideal age for rapid development of basic swimming skills is considered to be between 5 and 8 years of age [15,27,28]. In Poland, swimming classes begin at the age of 9 years, which is due to the organization of the educational system, many years of experience in this field, and social aspects, and above all the dynamic changes in morphological, motor, intellectual and social development, the harmony of the development, and specific motor excellence of children causing the effects of learning to swim in this period of life to be generally more pronounced than earlier.

Gender does not play a major role in the course of learning to swim and its outcomes in girls and boys aged 9–10 years, although in the group of those learning from scratch, boys acquire new swimming skills faster than girls [26]. The small gender differences in the rate of learning to swim are primarily due to the still small differences in body composition between girls and boys, who, in this period of life, are at the so-called golden age of motor abilities before puberty.

According to many researchers, the biggest problem in the first stage of teaching and learning to swim is the fear of water, caused by motor limitations, difficulty in maintaining balance, and a sense of threat to health, well-being, or life [16,20,29,30,31,32,33]. Fear of water is considered the strongest predictor of no or little progress in learning to swim [22,34,35]. 

Morphological characteristics play an equally important role in learning to swim, especially in the early stages [6,36,37]. Short individuals with strong and stocky bodies make more progress in the early years of learning, while tall and slim children perform better in the subsequent stages of swimming improvement and swimming training [38]. In contrast to overweight and obese individuals, slim peers have less body buoyancy and experience greater thermal discomfort. Initially, these factors make learning to swim easier for the former and more difficult for the latter. In later stages, when increasingly difficult motor tasks are involved, motor abilities limited due to obesity can become a major barrier to progress [39]. Flexibility is also beneficial in learning to swim. Good flexibility, especially in girls, enables faster development of correct swimming technique [21].

Faster learning to swim is also facilitated by a larger lung capacity, with its temporary increase after breathing in and holding breath increasing the buoyant force acting on the body, making it easier to stay afloat in a horizontal position and perform paddling movements. The student’s belief that it is easier to stay afloat after holding breath, even in stillness, is conducive to removing the fear of water, making it easier to begin learning dynamic swimming [40].

In learning to swim, coordination motor abilities, i.e., movements that are subordinated (ordered, concerted, and harmonized) to the goal to be achieved, are considered by many researchers to be more important than basic morphological and functional characteristics [19]. A higher level of coordination enables more accurate learning of the swimming technique [41]. Coordination motor abilities in swimming should be considered from the perspective of their importance for correct movements of the upper and lower limbs, coordination of large muscle groups of the upper and lower limbs and the whole body, synchronization of movements with breathing, overcoming external resistance created by water, and the so-called water feeling [42]. In general, boys exhibit greater explosive strength, coordination, and endurance, while girls are more flexible and able to perform motor activities with greater frequency. However, at the ages of 6–8 and 10–12 years, girls outperform boys in most characteristics [21].

To the best of our knowledge, the vast majority of studies that have assessed learning outcomes looked for factors of progress. However, there is a noticeable paucity of studies in which researchers attempt to address students who have struggled to learn to swim. Therefore, the purpose of the present study was to investigate which, and to what extent, selected personal determinants of children aged 9–10 years may limit the final achievement level in learning to swim. In view of the above, the research hypothesis was formulated, assuming that some children, despite regular attendance at swimming classes, do not achieve the learning outcomes set in the curriculum. The reason for this may be unfavorable (in relation to their peers) values of morphological and functional characteristics, coordination motor abilities, and problems with fear of water.

## 2. Materials and Methods

The examinations were conducted during the school year, from September to June of the following school year, at the swimming pool of the University of Physical Education in Krakow, Poland, on a group of 10-year-old students attending third grades of elementary schools. At the beginning of the school year, 540 non-swimmers were chosen from all children participating in the swimming program (981 students). Of this group, 271 children (137 girls and 134 boys) who had taken at least 25 swimming lessons during the school year and performed swimming tests and were measured for their somatic and functional characteristics and coordination motor abilities, were qualified for the examinations and analysis after the end of the school year. The level of swimming skills was evaluated based on interviews. The experiment was approved by the Bioethics Committee at the Regional Medical Chamber (No. 309/KBL/OIL/2019).

Swimming instruction was conducted based on a program developed by the Cracow School Sports Center, during the school year, once a week for 45 min. The program included adaptation to the aquatic environment (lessons 1–6), teaching swimming using basic techniques (lessons 7–13), teaching backstroke swimming (lessons 14–22), teaching front crawl swimming (lessons 23–33), and swimming tests (lessons 34–35). A detailed description of the swimming program is presented in Appendix A.

### 2.1. Procedure of Swimming Tests

The swimming tests developed by the authors, based on direct observation using an observation sheet and a scoring system, were used to measure the swimming skills of children aged 10 years.

The swimming tests, depending on the skills, included 1 to 3 tests. In the first test, the participant was asked to demonstrate skills related to adaptation to the aquatic environment. If the result was positive, the participant proceeded to the second test (backstroke swimming), and, if he or she swam the set distance using backstroke, they participated in the third test (front crawl swimming).

To assess the initial adaptation to the aquatic environment, the swimming test was conducted in the shallow water zone and consisted in performing a front glide over a distance of up to 7 m, whereas to measure swimming skills in backstroke and front crawl, the participants swam a distance of 15 to 25 m in a predefined manner. A distance of up to 7 m was considered sufficient to measure skills related to initial adaptation to the aquatic environment, and a distance of more than 15 m was considered sufficient to measure backstroke and front crawl swimming skills.

Swimming skills, indicative of progress in swimming, were assessed in points that reflected the swimming technique presented by the participants. Front glide, indicating initial adaptation to the aquatic environment, was scored 0 to 3 points, backstroke swimming with paddling movements of the lower limbs or with paddling movements of the upper and lower limbs was scored 0 to 6 points, and front crawl swimming—0 to 9 points. Therefore, the student could score between 0 and 18 points.

In the school year prior to the main research, a pilot study was conducted on third-grade elementary school students to determine the reliability of the swimming skill assessment tools. The reliability of the applied swimming tests was evaluated using Spearman’s rho correlation analysis with the following variants:evaluation of a student by two independent experts at the same time,expert evaluation with simultaneous video recording,expert evaluation performed twice, one week apart [43].

The direct observations of swimming skills met the reliability requirements, as the expert’s ratings were significantly correlated with the second expert’s rating or with the video, and the correlation coefficient was very high (0.81 to 0.91). Scoring details are presented in Appendix A.

### 2.2. Measuring Fear of Water with a Pre-Test

The Fear of Water Test was used to assess fear of water in children who could not swim. Its design was based on the Criterion Test of Anxiety [44].

In the pre-test, fear of water was measured based on direct observation using an observation sheet and a scoring system of the child’s behavior during climbing down the ladder into the water and walking on the outside lane of the swimming pool, in waist-to-chest water over a distance of 5 m, after which the child was classified according to the adopted criteria into a group:no fear of water (0 points): Unattended performance of the task,average fear of water (1 point): Performing the task in a focused and careful manner near the edge of the pool,high fear of water (2 points): Refusing to perform a task or performing a task while clinging to the edge of the pool.

### 2.3. Measuring Fear of Water with a Post-Test

The measurement of fear of water during the post-test was carried out in the shallow water zone, from waist to chest deep, on the outside lane of the swimming pool while performing a front glide over a distance of up to 7 m, after which children were classified according to the adopted criteria into the following groups:no fear of water (0 points): Unattended performance of a front glide with legs moving alternately and face in the water,average fear of water (1 point): Performing a front glide with legs moving alternately and face in the water but for too short a time (about 1–2 s), and making unnecessary nervous movements,high fear of water (2 points): Refusing to perform a glide or aborting a front glide during submerging the face in the water and pulling the legs off the bottom.

In the school year prior to the main research, a pilot study was conducted to assess the reliability of the Fear of Water Test. The test was conducted during the first classes among all children included in the swimming program. The reliability of the Fear of Water Test was evaluated using Spearman’s rho correlation analysis with the following variants:evaluation of the fear of water by two independent experts at the same time,expert evaluation with simultaneous video recording,evaluation performed twice, one week apart [43].

The direct observations of fear of water met the reliability requirements, as the expert’s ratings were significantly correlated with the second expert’s rating or with the video, and the correlation coefficient was very high (0.81 to 0.89).

### 2.4. Measurement of Basic Morphological and Functional Characteristics

Single tests of basic morphological characteristics and selected functional traits were carried out strictly according to the instructions, in the order in which the testing equipment was arranged, and concerned:body height and body mass,trunk flexibility according to Eurofit [45],foot mobility [38],vital capacity (VC) [46].

An anthropometer was used to measure body height, with an accuracy of 0.5 cm, while body weight was determined using an electronic balance with an accuracy of 0.1 kg. Body height was assessed using a stadiometer (Seca 213, Seca GmbH & Co, Hamburg, Germany) with a precision of 0.5 cm, while the body mass and its composition were determined by the method of electrical impedance measurement using the InBody 220 device (Biospace Co., Tokyo, Japan).

The child entered the examination in sports clothes and barefoot. Based on body height and weight, BMI was calculated using the formula: BMI = body weight (kg)/body height^2^ (m) (WHO). Subcutaneous fat was also measured using a Holtain-type skinfold caliper with a contact force of 10 g/mm2 for the triceps skinfold, subscapular skinfold, and supraspinale skinfold, with an accuracy of 0.1 mm.

### 2.5. Measurement of Coordination Motor Abilities

Among many tests assessing physical fitness which may have an impact on learning to swim of 10-year old children, we chose those that allow for the assessment of the efficiency of the nervous system at different levels of sensorimotor coordination (at a lower level: Reaction time; at a higher level: Visual-motor coordination and spatial orientation). The tests evaluated the following parameters:
reaction time to visual and auditory stimuli: This parameter was evaluated using an MRK-80 m; the participant performed 5 tests of reaction time to a visual stimulus and 5 to an auditory stimulus; from each series of measurements, the best and the worst results were rejected, and, using the remaining results, the mean reaction time (in ms) was calculated,visual-motor coordination—By means of Piórkowski US-9 apparatus, using the rate of stimuli presented appropriate for children aged 10 years, i.e., 93 stimuli per minute, the time of correctly received stimuli and the number of errors were measured; the number of correct answers was recorded,spatial orientation—using an AKN-102 cross apparatus; the rate of stimulus emission forced by the participant was applied; in the free series (without an imposed rhythm), the time [s] of correct responses to 49 stimuli was recorded [47].

### 2.6. Statistical Analysis

The results obtained in the study were processed statistically. For continuous data, mean values and their corresponding standard deviations were calculated, while numerical and percentage values were given for categorical data. Furthermore, coefficients of variation and minimum and maximum values were determined.

Spearman’s rank correlation coefficient was used to determine the relationship between variables describing the fear of water, morphological and functional characteristics, coordination motor abilities, and final achievement level in swimming of 10-year-old children. Pearson chi-square test was used to evaluate the correlations between the gender of the participants and their achievement levels. The hypothesis was verified at a significance level of α = 0.05 [48].

The participants were divided into groups according to achievement levels in swimming assessed based on swimming skills determined by the number of points. A t-test for independent samples and its non-parametric counterpart, the Mann-Whitney U-test, were used to determine differences between groups of achievement levels for each indicator.

## 3. Results

Based on the scores of the 10-year-old children on the end-of-year swimming tests, 5 swimming progress groups were formed, as shown in Table 1.

The most numerous groups were those of large (27.8%), average (26.3%), and little progress (24.1%). No progress was found in 6.3% of the participants, and 15.6% reported minimal progress in swimming (Table 1).

For the purposes of the present study, two groups were formed using the distribution presented in Table 1:a group with achievement level below the requirements set out in the school’s curriculum, which included children who scored 0–6 on the end-of-year tests, i.e., with no progress, minimal progress, or little progress in swimming,a group with the achievement level meeting the objectives set out in the school’s curriculum, which included children who scored 7–18 on the end-of-year tests, i.e., those with moderate to high progress in swimming. The results for each group are shown in Table 2.

Students whose participation in structured swimming lessons resulted in achievement levels meeting the requirements of the curriculum accounted for 53.9% of the participants. Others (46.1%) were classified as below the expected level. Based on the chi-square test, there was no significant relationship between gender and swimming achievement levels (*p* = 0.525).

### 3.1. Relationships between Fear of Water, Morphological Characteristics, Functional Traits, Coordination Motor Abilities, and Progress in Swimming

Relationships between fear of water, morphological characteristics, functional characteristics, coordination motor abilities of girls and boys, and achievement level in swimming (Spearman’s rho correlation coefficients) are shown in Figure 1.

From Figure 1 above, it can be concluded that there was a multiple effect of the factors studied. The highest values of Spearman’s rho correlation coefficient were found for fear of water and spatial orientation with final achievement levels, while the lowest values, which were not statistically significant, were observed for trunk flexibility, foot mobility, the sum of three skinfolds, and reaction time to an auditory stimulus. It is noteworthy that morphological characteristics and coordination motor abilities had a greater effect on achievement level in swimming in boys, whereas flexibility and fear of water affected more the group of girls.

### 3.2. Effect of Fear of Water on Progress in Swimming

The level of fear of water measured on the pre-test determined learning new skills by children and the related achievement levels. The results of the study according to individual groups of girls and boys are presented in Table 3.

Based on the analyses presented in Table 3, it was found that students whose achievement level in swimming was below the requirements of the curriculum had higher levels of fear of water tested on the pre-test than the group of children who progressed to meet the requirements. Higher levels of fear of water were found in girls in both study groups, especially in the group with progress below the requirements (*p* < 0.001).

### 3.3. Effect of Individual Morphological Characteristics on Progress in Swimming

Morphological characteristics that may be determinants of the achievement level in swimming include body height, body weight, body fatness expressed by the sum of the three skinfolds, and BMI. The results are shown in Table 4.

In all cases related to selected morphological characteristics, lower achievement levels were observed in students with lower body height, lower body weight, a lower sum of three skinfolds, and lower BMI. Despite the differences, these values were statistically insignificant, except for body height in boys (*p* < 0.05).

### 3.4. Effect of Functional Characteristics on Progress in Swimming

Vital capacity, foot mobility, and trunk flexibility were considered to be functional characteristics that may be determinants of the achievement level in swimming. The results are shown in Table 5.

Of the functional characteristics, the greatest effect on achievement level in swimming was found for vital capacity. Among both girls and boys with achievements below the requirements of the curriculum, this characteristic was significantly lower (*p* < 0.01). In girls, foot mobility and trunk flexibility were lower in the group progressing below the requirements, but there were no statistically significant correlations. In boys, this trait was similar in both swimming progress groups.

### 3.5. Effect of Selected Coordination Motor Abilities on Progress in Swimming

Selected motor abilities that may be determinants of the achievement level in swimming include reaction time to visual stimulus, reaction time to an auditory stimulus, visual-motor coordination, and spatial orientation. The results are shown in Table 6.

The results of selected coordination motor abilities in both girls and boys were significantly different between the group with swimming progress below the requirements set out in the curriculum and progress meeting these objectives. In the group of girls, significant differences were found in all test results, especially in visual-motor coordination and spatial orientation (*p* < 0.001). In boys, there was no correlation between the reaction time to an auditory stimulus, while in the other tests, correlations were significant, especially in spatial orientation (*p* < 0.001).

## 4. Discussion

The level of progress in swimming depends on individual aptitudes and on the means and methods of teaching used in the process of learning to swim, and the expertise and personality of the teacher. The most significant individual factors determining the effectiveness of learning to swim are the level of physical development and motor abilities (especially coordination), fear of water, and previous experience. The age of learners and their associated intellectual development are also significant [15].

Participation in an adequate number of swimming lessons is considered fundamental to acquiring swimming skills [25,26]. A similar relationship was found in our study. Based on the analysis of the results, 93.7% of children participating in structured swimming lessons acquired new swimming skills, but to varying degrees. The minimum requirements of the school curricula concerning swimming were met by only 54.1% of the participants, of whom 26.3% learned backstroke swimming and 27.8% also learned front crawl. The remainder (45.9%) were students with achievement levels below the curriculum objectives, of which 6.3% failed to learn any water activity and 15.6% learned only activities related to initial adaptation to the aquatic environment. The differences between boys and girls were statistically insignificant at χ^2^ (4) = 3.20, *p* = 0.525, but more girls than boys were found in the groups of no progress and little progress. The small effect of gender at age 10 years on the final achievement level has also been reported by other researchers, claiming that at this age, there are negligible differences in body composition that determine learning new skills [24,25,26].

It is considered that the factors with the greatest effect on learning to swim, especially in the early stages, include age and the associated gender differences, physical, motor, mental, and social development, and environmental conditions, such as the teacher and the methods, forms, and means of teaching they use [2,19,20,21,22,23,25,26]. Our study mostly confirmed previous findings concerning the relationships of the physical development of 10-year-old children and their associated morphological and functional characteristics, coordination motor skills, and fear of water with the achievement level in swimming [2,19,20,21,22,23,25,26]. The highest values of Spearman’s rho correlation coefficient were found for fear of water and spatial orientation with swimming progress, while the lowest values, which were not statistically significant, were observed for trunk flexibility, foot mobility, the sum of three skinfolds, and reaction time to an auditory stimulus. It is noteworthy that morphological characteristics and coordination motor abilities had a greater effect on the achievement level in boys, whereas flexibility and fear of water affected more the group of girls.

The relationships of fear of water, morphological and functional characteristics, and coordination abilities of girls and boys with the achievement level in swimming are even more pronounced when a comparison is made between children with the achievement level meeting the requirements set out by the school’s swimming curriculum and children with this level below the curriculum objectives. In non-swimmers, the fear of losing balance in the water, flooding the face, eyes, choking, and having to pull the legs off the ground is quite a problem [20,30,31,32,33], which was confirmed by our findings. Students who performed worse in learning to swim were characterized by significantly higher fear of water than higher-performing children. Higher levels of fear of water were found in girls, especially in the group with an achievement level below curriculum objectives (*p* < 0.001). This may mean that fear of water was a stronger determinant of learning to swim in girls than in boys.

In learning to swim, especially in the early stages, an equally important role has been attributed to morphological characteristics and body physique and composition [6,38]. The results indicated that in all cases related to selected morphological characteristics, lower achievement levels in swimming occurred in children characterized by both lower body height and body weight, a lower sum of three skinfolds, and lower BMI. The differences in somatic characteristics between the progress groups studied were not statistically significant, except for body height in boys (*p* < 0.05). It can be presumed that individuals who were shorter and with lower body mass at the same time, characterized by low body fat, had more reasons for fear, even in the shallow water zone, because it could seem too deep for them. Low body mass and the associated low body fatness is another barrier due to poorer body buoyancy, making it difficult to perform water buoyancy familiarization exercises, and causing more rapid hypothermia that discourages water exercise.

Faster learning to swim in the first stage is favored by better body buoyancy, which is related, among other things, to higher vital capacity [49]. Having more breathing capacity may be conducive to the student’s belief that they will stay afloat on the surface of the water more easily and longer after holding their breath. In the first stage of learning, good body buoyancy compensates for skill deficiencies, whereas the person’s confidence in the ability to stay afloat promotes a quicker overcoming of the fear of water, making it easier to start learning dynamic swimming. In the case of poorer buoyancy, in order to stay afloat in a horizontal position, the learner has to perform more intensive paddling movements or use buoyancy aids, whereas the deeper immersion of the body and the associated filling up the mouth with water causes the learner to stop exercising while discouraging further attempts. If this is the case, progress in learning to swim may be impeded [40]. The results obtained in the study confirmed this relationship. Individuals with lower vital capacity had lower achievement levels and thus made significantly less progress in swimming (*p* < 0.01).

Some researchers [21] found that learning to swim is facilitated by good flexibility, especially in girls, which enables faster development of correct swimming technique. Our study failed to confirm this relationship. Although foot and trunk mobility in girls in the group of progress below the requirements set out in the curriculum were lower than in the group with the progress meeting the curriculum objectives, there were no significant intergroup correlations, whereas in boys, these characteristics were similar in both progress groups. It can be speculated that they may play a more significant role in the subsequent stages of development of swimming skills, i.e., skill improvement and sports training.

In learning to swim, a child’s coordination motor abilities, i.e., movements that are subordinated (ordered, concerted, and harmonized) to the goal to be achieved, are considered to be more important than basic morphological and functional characteristics [19]. A higher level of coordination enables learning of a more accurate technique associated with correct upper and lower limb movements, joint work of large muscle groups, synchronization of movements with breathing, and water sensation [21,41,50]. Our study confirmed these relationships in the progress groups of children beginning to learn to swim from scratch. On almost all tests determining coordination motor abilities, children with swimming progress below the requirements of the school’s curriculum performed significantly worse than children in the group of final achievement levels meeting the curriculum objectives. Among girls, these correlations occurred especially in visual-motor coordination and spatial orientation (*p* < 0.001), whereas in boys—in spatial orientation (*p* < 0.001).

Analysis of the relationships of anxiety, morphological and functional characteristics, and coordination motor abilities with progress in swimming reveals a mutual negative or positive interaction of the characteristics studied. The children’s fear of water may have been related to their poorer body buoyancy due to lower body height, body weight, body fatness, BMI, and thus lower vital capacity. Perhaps lower values of somatic characteristics correlate with poorer or delayed physical development, which may have resulted in poorer performance, especially in coordinative motor abilities. The sum of favorable individual determinants may have accelerated the progress in learning to swim in many cases, while unfavorable determinants may have been a significant barrier. The above doubts should be further addressed in future research in order to broaden the knowledge on the problems discussed in the study. It is also necessary to strive to search for other factors that may affect the progress in learning to swim.

## 5. Conclusions

In many cases, children who begin learning to swim from scratch make significant progress. However, for many of them, the achievement level is lower than the requirements set out in the school curriculum. The biggest problem for teachers and students in the initial teaching and learning to swim was the high fear of water, especially among girls. Children characterized by lower body height and weight, a lower sum of the three skinfolds, lower BMI, and lower vital capacity also experience problems with progress in swimming. However, a significant role should not be attributed to flexibility in the first stage of learning to swim, in contrast to coordination motor abilities, which are significantly worse in those with swimming achievement levels below the requirements set out in the curriculum compared to children in the group with this level meeting these objectives.

## Figures and Tables

**Figure 1 ijerph-19-05663-f001:**
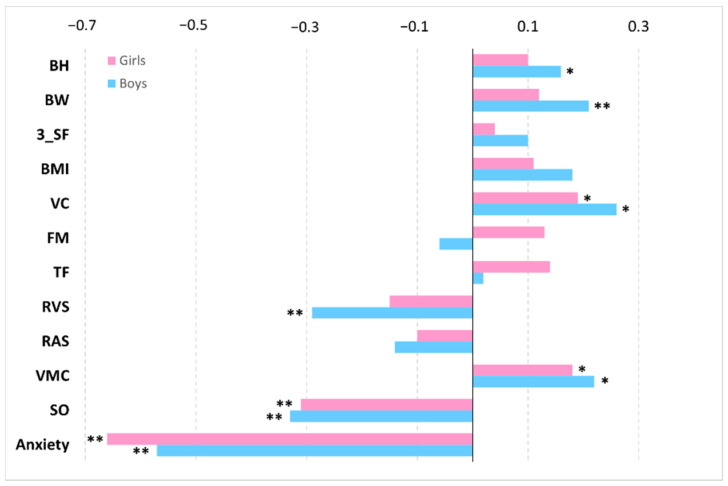
Relationships between fear of water, morphological characteristics, coordination motor abilities of girls and boys, and achievement level in swimming (Spearman’s rho correlation coefficients). Notes: BH—body height, BW—body weight, 3 SF—sum of three skinfolds, BMI—Body Mass Index, VC—vital capacity, FM—foot mobility, TF—trunk flexibility, RVS—reaction time to visual stimulus, RAS—reaction time to an auditory stimulus, VMC—visual-motor coordination, SO—spatial orientation. ^⁎^
*p* < 0.05; ^⁎⁎^
*p* < 0.01.

**Table 1 ijerph-19-05663-t001:** Groups of progress in swimming for 10-year-old children based on skills expressed in points obtained in the final tests.

Progress	Gender				Total	
	Girls		Boys		N	%
	N	%	N	%		
No progress (0 points)	12	8.8	5	3.8	17	6.3
Minimal progress (1–3 points)	20	14.6	22	16.5	42	15.6
Little progress (4–6 points)	34	24.8	32	23.3	66	24.1
Average progress (7–11 points)	34	24.8	36	27.8	70	26.3
Great progress (12–18 points)	37	27.0	39	28.6	76	27.8
Total	137	100.0	134	100.0	271	100.0

**Table 2 ijerph-19-05663-t002:** Characteristics of 10-year-old children in swimming progress groups.

Progress	Girls	Boys	Total
	N %	N %	N %
Below the requirements (0–6 points)	66 48.2	59 44.0	125 46.1
Meeting the requirements (7–18 pts)	71 51.8	75 56.0	146 53.9
Total	137 100.0	134 100.0	271 100.0

χ^2^(4) = 3.20; *p* = 0.525.

**Table 3 ijerph-19-05663-t003:** Fear of water in 10-year-old children in swimming progress groups.

Characteristic	Progress	Girls			Boys		
		m ± sd	v	Min–Max	m ± sd	v	Min–Max
Fear of water	Below the requirements	1.75 ± 0.4 ^⁎⁎⁎^	24.4	0–2	1.37 ± 0.7 ^⁎^	50.0	0–2
	Meeting the requirements	1.18 ± 0.6	53.7	0–2	1.09 ± 0.6	58.5	0–2

^⁎^*p* < 0.05; ^⁎⁎⁎^
*p* < 0.001.

**Table 4 ijerph-19-05663-t004:** Morphological characteristics of 10-year-old children in swimming progress groups.

Characteristic	Progress	Girls			Boys		
		m ± sd	v	Min–Max	m ± sd	v	Min–Max
Body height	Below the requirements	138.0 ± 7.7	5.6	123–160	138.1 ± 6.6 ^⁎⁎^	4.8	120–153
	Meeting the requirements	139.7 ± 7.2	5.2	121–158	141.1 ± 6.2	4.4	128–158
Body weight	Below the requirements	32.9 ± 7.0	21.2	22.1–56.3	33.2 ± 7.0	20.9	19.7–53.5
	Meeting the requirements	34.6 ± 7.5	21.8	24.1–59.8	35.7 ± 7.2	20.2	20.5–72.1
Sum of three skinfolds	Below the requirements	36.6 ± 13.0	34.9	19.6–75.3	32.6 ± 15.9	47.8	17.5–91.0
	Meeting the requirements	37.6 ± 15.2	40.4	14.8–87.0	34.3 ± 14.5	42.4	15.0–85.5
BMI	Below the requirements	17.1 ± 2.3	13.0	13.4–26.4	17.7 ± 2.9	16.8	13.6–26.9
	Meeting the requirements	17.6 ± 2.6	15.0	13.9–26.6	17.8 ± 2.6	14.8	11.5–28.9

^⁎⁎^*p* < 0.01

**Table 5 ijerph-19-05663-t005:** Functional characteristics of 10-year-old children in swimming progress groups.

Characteristic	Progress	Girls			Boys		
		m ± sd	v	Min–Max	m ± sd	v	Min–Max
VC	Below the requirements	2052 ± 460 ^⁎⁎^	22.5	1100–3400	2288 ± 414 ^⁎⁎^	17.9	1300–3100
	Meeting the requirements	2264 ± 441	19.4	1500–3800	2530 ± 415	16.4	1500–3700
RS	Below the requirements	73.3 ± 10.14	13.9	58–95	74.2 ± 9.7	13.3	53–93
	Meeting the requirements	76.1 ± 11.63	15.3	57–110	74.8 ± 11.0	14.7	49–99
GT	Below the requirements	64.4 ± 8.67	13.5	45–78	62.3 ± 7.5	11.7	49–76
	Meeting the requirements	66.8 ± 7.48	11.1	51–84	61.9 ± 7.6	17.5	41–77

Notes: VC—vital capacity, RS—foot mobility, GT—trunk flexibility, ^⁎⁎^
*p* < 0.01

**Table 6 ijerph-19-05663-t006:** Coordination motor abilities in 10-year-old children in swimming progress groups.

Characteristic	Progress	Girls			Boys		
		m ± sd	v	Min–Max	m ± sd	v	Min–Max
SWR	Below the requirements	291.9 ± 45.4 ^⁎⁎^	15.4	200–437	290.3 ± 57.0 ^⁎⁎^	19.2	213–566
	Meeting the requirements	271.1 ± 36.8	13.6	193–386	264.7 ± 35.4	13.4	206–440
SRS	Below the requirements	281.7 ± 52.3 ^⁎^	18.5	200–453	273.0 ± 60.5	22.1	126–380
	Meeting the requirements	264.8 ± 33.2	12.5	200–333	260.6 ± 34.4	13.2	193–426
KWR	Below the requirements	56.5 ± 18.3 ^⁎⁎⁎^	35.5	38–88	54.2 ± 19.0 ^⁎⁎^	36.2	36–84
	Meeting the requirements	66.8 ± 14.4	21.6	33–89	63.0 ± 16.0	25.7	36–96
OP	Below the requirements	126.2± 29.2 ^⁎⁎⁎^	22.7	78–252	124.2± 35.0 ^⁎⁎⁎^	27.7	65–286
	Meeting the requirements	105.1 ± 18.6	17.7	71–167	104.6 ± 20.4	19.3	68–207

Notes: SRW—reaction time to visual stimulus [ms], SRS—reaction time to auditory stimulus [ms], KWR—visual-motor coordination [number of correct answers], OP—spatial orientation [s], ^⁎^
*p* < 0.05; ^⁎⁎^
*p* < 0.01; ^⁎⁎⁎^
*p* < 0.001.

## Data Availability

The data presented in this study are available on request from the corresponding author.

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
