# Peer review of "Individual Determinants as the Causes of Failure in Learning to Swim with the Example of 10-Year-Old Children"

_ijerph, 2022, doi:10.3390/ijerph19095663_

Round 1
Reviewer 1 Report
I commend the authors on undertaking this work, as whilst swimming lessons are recognised as an important strategy for drowning prevention, little is known about how individual determinants impact on the acquisition of swimming skills. The manuscript was a pleasure to read, I have just provided some comments below that I hope will assist in enhancing the quality of the paper.
Abstract:
* The conclusion of the abstract could be strengthened to better align with the study aim and add substance to the abstract.
Materials and Method:
- Further justification/explanation to more fully describe how participants were selected. For example, was this sample inclusive of all non-swimmers?
- How did the authors define a non-swimmer for the purpose of this manuscript? I would recommend that the authors consider including this within the manuscript.
- The manuscript would be strengthened if additional details of the swimming program were provided? It might be possible for this to be articulated within a Table? This would enhance the capacity for replication of the program/study with different age groups and populations.
Procedure of Swimming Tests
- Additional detail is required to explain the scoring system. For example, how did one differentiate between scores?
Results
Formatting of results heading requires correction
Discussion
Some repetition in text between the discussion and the introduction, for example, p 10 lines 397 to 402.
Wishing you all the best with the next iteration of this manuscript and looking forward to seeing the published version.
Author Response
Dear Reviewer,
Thank you very much for your time and valuable comments, which all have been considered and incorporated. The detailed list of responses is given below. We hope that the modifications and explanation will be acceptable for you.
Yours sincerely,
Rydzik, corresponding author
I commend the authors on undertaking this work, as whilst swimming lessons are recognised as an important strategy for drowning prevention, little is known about how individual determinants impact on the acquisition of swimming skills. The manuscript was a pleasure to read, I have just provided some comments below that I hope will assist in enhancing the quality of the paper.
Abstract:
* The conclusion of the abstract could be strengthened to better align with the study aim and add substance to the abstract.
A: This part has been corrected
Materials and Method:
- Further justification/explanation to more fully describe how participants were selected. For example, was this sample inclusive of all non-swimmers?
A: The sample included all non-swimmers.
- How did the authors define a non-swimmer for the purpose of this manuscript? I would recommend that the authors consider including this within the manuscript.
A: Non-swimmers were verified through interviews. The relevant information has been added to the manuscript.
- The manuscript would be strengthened if additional details of the swimming program were provided? It might be possible for this to be articulated within a Table? This would enhance the capacity for replication of the program/study with different age groups and populations.
A: Due to the large volume, the information has been added in supplementary materials.
Procedure of Swimming Tests
- Additional detail is required to explain the scoring system. For example, how did one differentiate between scores?
A: Due to the large volume, the information has been added in supplementary materials.
Results
Formatting of results heading requires correction
A: This part has been corrected
Discussion
Some repetition in text between the discussion and the introduction, for example, p 10 lines 397 to 402.
A: This part has been corrected
Wishing you all the best with the next iteration of this manuscript and looking forward to seeing the published version
Reviewer 2 Report
To the Authors, You have addressed an important issue, i.e. what factors affect learning to swim among 9-10 yr old children. A considerable amount of data has been collected and treated statistically in an appropriate way. The selection of the working hypothesis is relevant. Well done!
The article however, has several weaknesses. The most important may be that it was not edited by an English speaker. In some cases the meaning of the text is unclear. It is difficult to say whether this is the result of the failure to edit by an English speaker or whether there may in fact, be some inexact information/statements. Some examples of unclear statements or aims are:
- Progress was considered to be in relation to that which is in the school curriculum. No explanation of what is in the curriculum is included. The authors assume that it is appropriate. This may not be the case.
- Use of the word "progress" suggests addressing the "rate" of learning. "Fast progress" even more so. This does not seem to have been done. The word progress seems to be used in reference to final achievement level.
- The subjects were tested for their skill accomplishment after the intervention. They were not tested before. It is thus difficult to know how much they really achieved as a result of the intervention. Several previous studies have suggested that perhaps the primary factor affecting learning rate or final level of accomplishment is prior experience and especially the level of skill at the start. This reviewer considers this a major omission which cannot be corrected after the study has been completed (unless this data is somehow available in your records).
- The subjects were judged non-swimmers prior to intervention. No criteria are described with which this designation was determined.
- Fear is cited as the greatest inhibitor of learning. This reviewer feels that the criteria used to describe fear are inadequate. Furthermore, the impression is given that fear was common among the learners. This may be a semantic issue. No attempt is made to distinguish between being healthily and normally cautious and actual fear. In this reviewers experience, fear is not common.
Other issues that may be semantic in nature are, if not simply inexact, are:
a. Movement is said to affect buoyancy - which of course it does not. See line 47.
b. Floating capacity is several times referred to as floating in a horizontal position. Floating capacity and floating angle are theoretically unrelated. A person can have low density (high floating capacity) yet still have a low passive floating angle.
c. The bodies "buoyant force" is referred to. The force of course is not in the body but in the water. Again, possibly a language issue.
I have recommended to the editors that the article should be accepted with revisions. The primary revision is editing by an English speaker who is familiar with the issues involved in the study. Whether this is minor or major I am unable to judge. The language issues cited above are minor but can be misleading. You risk being misunderstood.
Otherwise I anticipate an improved, relevant and important article to eventually appear. Again, well done and thank you.
Author Response
Dear Reviewer,
Thank you very much for your time and valuable comments, which all have been considered and incorporated. The detailed list of responses is given below. We hope that the modifications and explanation will be acceptable for you.
Yours sincerely,
Rydzik, corresponding author
To the Authors, You have addressed an important issue, i.e. what factors affect learning to swim among 9-10 yr old children. A considerable amount of data has been collected and treated statistically in an appropriate way. The selection of the working hypothesis is relevant. Well done!
The article however, has several weaknesses. The most important may be that it was not edited by an English speaker. In some cases the meaning of the text is unclear. It is difficult to say whether this is the result of the failure to edit by an English speaker or whether there may in fact, be some inexact information/statements. Some examples of unclear statements or aims are:
A: English was corrected by a native speaker
- Progress was considered to be in relation to that which is in the school curriculum. No explanation of what is in the curriculum is included. The authors assume that it is appropriate. This may not be the case.
A: The curriculum has been added in the appendix.
- Use of the word "progress" suggests addressing the "rate" of learning. "Fast progress" even more so. This does not seem to have been done. The word progress seems to be used in reference to final achievement level.
A: This has been corrected
- The subjects were tested for their skill accomplishment after the intervention. They were not tested before. It is thus difficult to know how much they really achieved as a result of the intervention. Several previous studies have suggested that perhaps the primary factor affecting learning rate or final level of accomplishment is prior experience and especially the level of skill at the start. This reviewer considers this a major omission which cannot be corrected after the study has been completed (unless this data is somehow available in your records).
A: Dear Reviewer, thank you for your comment. We would like to point out that the study was conducted on subjects who were initially unable to swim and therefore it was impossible to conduct a prior swimming test.
- The subjects were judged non-swimmers prior to intervention. No criteria are described with which this designation was determined.
A: Relevant information has been added. Swimming skills were verified by interviews.
- Fear is cited as the greatest inhibitor of learning. This reviewer feels that the criteria used to describe fear are inadequate. Furthermore, the impression is given that fear was common among the learners. This may be a semantic issue. No attempt is made to distinguish between being healthily and normally cautious and actual fear. In this reviewers experience, fear is not common.
A: The inhibitory factor was fear of water, but other inhibitors such as morphological structure also appeared. Slim people had trouble staying afloat which discouraged them from learning regularly; they also got cold quicker. It follows that fear was one of the factor, not the only one.
Other issues that may be semantic in nature are, if not simply inexact, are:
- Movement is said to affect buoyancy - which of course it does not. See line 47.
A: This has been corrected
- Floating capacity is several times referred to as floating in a horizontal position. Floating capacity and floating angle are theoretically unrelated. A person can have low density (high floating capacity) yet still have a low passive floating angle.
A: This has been corrected
- The bodies "buoyant force" is referred to. The force of course is not in the body but in the water. Again, possibly a language issue.
A: This has been corrected
I have recommended to the editors that the article should be accepted with revisions. The primary revision is editing by an English speaker who is familiar with the issues involved in the study. Whether this is minor or major I am unable to judge. The language issues cited above are minor but can be misleading. You risk being misunderstood.
Otherwise I anticipate an improved, relevant and important article to eventually appear. Again, well done and thank you.
A: Thank you for your kind words and your valuable comments